# On the Effectiveness of Adversarial Training Against Common Corruptions

**Klim Kireev**[*1]      **Maksym Andriushchenko**[*1]      **Nicolas Flammarion**[1]

[1]EPFL, Lausanne, Switzerland

## Abstract

The literature on robustness towards common corruptions shows no consensus on whether adversarial training can improve the performance in this setting. First, we show that, when used with an appropriately selected perturbation radius, $\ell_p$ adversarial training can serve as a strong baseline against common corruptions improving both accuracy and calibration. Then we explain why adversarial training performs better than data augmentation with simple Gaussian noise which has been observed to be a meaningful baseline on common corruptions. Related to this, we identify the *σ-overfitting* phenomenon when Gaussian augmentation overfits to a particular standard deviation used for training which has a significant detrimental effect on common corruption accuracy. We discuss how to alleviate this problem and then how to further enhance $\ell_p$ adversarial training by introducing an *efficient relaxation* of adversarial training with *learned perceptual image patch similarity* as the distance metric. Through experiments on CIFAR-10 and ImageNet-100, we show that our approach does not only improve the $\ell_p$ adversarial training baseline but also has cumulative gains with data augmentation methods such as AugMix, DeepAugment, ANT, and SIN, leading to state-of-the-art performance on common corruptions. The code of our experiments is publicly available at `https://github.com/tml-epfl/adv-training-corruptions`.

## 1 INTRODUCTION

Despite achieving human-level performance on many computer vision tasks, deep neural networks are still not as ro-

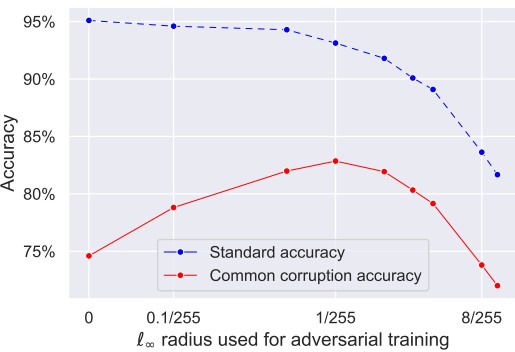

Figure 1: Accuracy on common corruptions from CIFAR-10-C for ResNet-18 models adversarially trained using different $\ell_\infty$ radii. We observe that the performance with $\varepsilon = 1/255$ is significantly higher than with the standardly used $\varepsilon = 8/255$.

bust as humans towards various distribution shifts [Szegedy et al., 2014, Taori et al., 2020] including common image corruptions [Hendrycks and Dietterich, 2019]. Attempts to understand the vulnerability towards such shifts include analysis of the network architecture [Azulay and Weiss, 2019], the features contained in the data [Ilyas et al., 2019], and frequency analysis of neural networks [Yin et al., 2019, Ortiz-Jimenez et al., 2020]. Many approaches have been suggested to improve their robustness to these shifts including approaches based on data augmentations [Cubuk et al., 2019, Hendrycks et al., 2019b], adversarial training [Madry et al., 2018, Laidlaw et al., 2021], and pretraining [Hendrycks et al., 2019a].

Although data augmentation methods tend to improve the performance under common synthetic corruptions [Hendrycks et al., 2019b], these augmentations are often ad hoc and may have substantial overlap with the corruptions evaluated at test time. At the same time, there is a large amount of literature on adversarial training with $\ell_p$-bounded perturbations [Goodfellow et al., 2015, Madry et al., 2018]. Adversarial training emerged as a principled approach to

---

[*]Equal contribution.

*Accepted for the 38th Conference on Uncertainty in Artificial Intelligence* (UAI 2022).

improve the worst-case performance of the model against *small* $\ell_p$ perturbations. However, common image corruptions have a very high $\ell_p$ distance from clean samples, so the utility of using $\ell_p$ adversarial training for them is not obvious. This leads us to explore the following question:

*How can we improve the performance on common image corruptions using adversarial training?*

We make the following contributions in our paper:

- We show that $\ell_p$ adversarial training with an *appropriately selected* perturbation radius can serve as a strong baseline against common image corruptions improving both accuracy and calibration on corrupted images.
- We analyze the success of $\ell_p$ adversarial training via a comparison to other natural baselines such as Gaussian data augmentation. We observe that it can overfit to the perturbation size it has been trained which, however, does not happen for adversarial training.
- We introduce an efficient relaxation of adversarial training with *learned perceptual image patch similarity* (LPIPS) [Zhang et al., 2018b] based on layerwise adversarial perturbations. This new relaxation is at least as effective as previous approaches [Laidlaw et al., 2021] but significantly faster to train.
- We show that our relaxation approach has cumulative gains with existing data augmentation methods such as AugMix, DeepAugment, ANT, and SIN leading to state-of-the-art performance on common corruptions from CIFAR-10-C and ImageNet-100-C.

## 2   RELATED WORK

We provide here an overview of relevant works on common image corruptions, different data augmentation methods proposed to improve the performance on corruptions, and then we discuss papers on adversarial robustness with respect to both $\ell_p$ and non-$\ell_p$ perturbations.

**Common image corruptions.** Dodge and Karam [2017] first find that despite being on par with the human vision on standard images, deep networks perform suboptimally on common corruptions such as noise and blur. Geirhos et al. [2018] measure the performance of deep networks on 12 different image corruption types but find that data augmentation on one type of corruption does not tend to improve the performance on others. However, these findings are reconsidered in Rusak et al. [2020] where Gaussian data augmentation is shown to help for a wide range of image corruptions. In a standardization effort, Hendrycks and Dietterich [2019] introduce a few image classification datasets—in particular, CIFAR-10-C and ImageNet-C—with 15 different common corruptions from four categories: noise, blur, weather, and digital corruptions. Ovadia et al. [2019] show

that not only acccuracy but also calibration deteriorates under these common corruptions. [Schneider et al., 2020, Nandy et al., 2021] show that robustness to common corruptions can be improved by using test-time adaptation, e.g., via recomputing the batch normalization statistics. Radford et al. [2021] show that contrastive pretraining on a very large set of image-caption pairs can substantially improve robustness on various distribution shifts including common corruptions.

**Data augmentations.** Data augmentation is a widely used technique to improve the generalization. Besides classical image transformations like random flipping or cropping, many other approaches have been proposed such as linearly interpolating between images and their labels [Zhang et al., 2018a], replacing a part of the image with either a black-colored patch [DeVries and Taylor, 2017] or a part of another image [Yun et al., 2019]. One of the best-performing methods in terms of accuracy and calibration on common corruptions is AugMix [Hendrycks et al., 2019b], which combines carefully selected augmentations with a regularization term based on the Jensen-Shannon divergence. Taori et al. [2020] observe that improvements on synthetic distribution shifts (such as common corruptions) do not necessarily transfer to real distribution shifts. However, Hendrycks et al. [2021] show an example when improving robustness against synthetic blurs also helps against naturally obtained blurred images.

$\ell_p$ **adversarial robustness.** Adversarial training in deep learning has been first considered in Goodfellow et al. [2015] and later framed as a robust optimization problem by Madry et al. [2018]. The view that adversarial training damages or at least does not improve the performance on *common corruptions* has been prevalent in the literature [Hendrycks et al., 2019b, Rusak et al., 2020, Hendrycks et al., 2021]. However, previous works directly use publicly available robust models without adjusting the perturbation radius used for adversarial training. For example, Rusak et al. [2020] show that adversarially trained ImageNet models from Xie et al. [2019], Shafahi et al. [2019], and Shafahi et al. [2020] do not help on ImageNet-C compared to standardly trained models. However, Ford et al. [2019] report that $\ell_\infty$ adversarially trained models on CIFAR-10 from Madry et al. [2018] do lead to an improvement on CIFAR-10-C compared to a standard model. The approach of Xie et al. [2020], AdvProp, relies on $\ell_\infty$ adversarial training to improve standard and corruption accuracy but they advocate the use of *auxiliary* batch normalization layers for standard and adversarial training examples. We find that similar performance can be achieved on common corruptions using vanilla adversarial training without a customized use of BatchNorm layers. Kang et al. [2019] study the robustness transfer between $\ell_p$-robust models and *adversarially optimized* elastic and JPEG corruptions. They show that $\ell_p$ adversarial training can increase robustness against these

two types of adversarial perturbations, but robustness does not transfer in all the cases and sometimes may even hurt robustness against other perturbation types.

**Non-$\ell_p$ adversarial robustness.** Volpi et al. [2018] propose Lagrangian-style adversarial training in the input space and in the last layer of the network. Stutz et al. [2019] propose *on-manifold* adversarial training which is performed in the latent space of a VAE-GAN generative model. However, its success crucially depends on the quality of the generative model which could not be scaled beyond simple image recognition datasets. Wei and Ma [2020] derive generalization bounds that motivate adversarial training with respect to all network layers which they use to improve $\ell_p$ robustness. Recently, Laidlaw et al. [2021] provided algorithms for approximate *perceptual adversarial training* based on the LPIPS distance [Zhang et al., 2018b] which is defined via activations of a neural network. They aim at improving robustness against new types of adversarial perturbations that were unseen during training.

# 3 $\ell_p$ ADVERSARIAL TRAINING IMPROVES THE PERFORMANCE ON COMMON CORRUPTIONS

Here we formally introduce adversarial training and show that it can lead to non-trivial improvements in accuracy and calibration on common corruptions.

**Background on adversarial training.** Let $\ell(x, y; \theta)$ denote the loss of a classifier parametrized by $\theta \in \mathbb{R}^m$ on the sample $(x, y) \sim D$ where $D$ is the data distribution. Previous works [Shaham et al., 2018, Madry et al., 2018] formalized the goal of training adversarially robust models as the following optimization problem:

$$\min_\theta \mathbb{E}_{(x,y) \sim D} \left[ \max_{\delta \in \Delta} \ell(x + \delta, y; \theta) \right]. \quad (1)$$

In this section, we focus on the $\ell_p$ threat model, i.e. $\Delta = \{\delta \in \mathbb{R}^d : \|\delta\|_p \le \varepsilon, \ x + \delta \in [0, 1]^d\}$, where the adversary can change each input $x$ in an $\varepsilon$-ball around it while making sure that the input $x + \delta$ does not exceed its natural range. A common way to solve the inner maximization problem is the *projected gradient descent* method (PGD) defined by the following recursion initialized at $\delta^{(0)}$:

$$\delta^{(t+1)} \stackrel{\text{def}}{=} \Pi_\Delta \left[ \delta^{(t)} + \alpha \nabla_{\delta^{(t)}} \ell(x + \delta^{(t)}, y; \theta) \right], \quad (2)$$

where $\Pi$ is the projection operator on the set $\Delta$, and $\alpha$ is the step size of PGD. Instead of the gradient, one often uses the gradient sign update for $\ell_\infty$ perturbations or the $\ell_2$ normalized update for $\ell_2$ perturbations. $\delta^{(0)}$ can be initialized as any point inside $\Delta$, e.g. as zero, or randomly [Madry et al., 2018].

The one-iteration variant of PGD is known as the *fast gradient method* (FGM) when the normalized $\ell_2$ update is used

Table 1: Accuracy and calibration of ResNet-18 models trained on CIFAR-10 and ImageNet-100. $\ell_\infty$ and $\ell_2$ adversarial training substantially improves accuracy and calibration error (ECE) on corrupted samples.

| Training | Standard accuracy | Corruption accuracy | Corruption calibration error |
|---|---|---|---|
| **CIFAR-10** | | | |
| Standard | 95.1% | 74.6% | 16.6% |
| $\ell_\infty$ adversarial | 93.3% | 82.7% | 10.8% |
| $\ell_2$ adversarial | 93.6% | **83.4%** | **10.5%** |
| **ImageNet-100** | | | |
| Standard | 86.6% | 47.5% | 10.0% |
| $\ell_\infty$ adversarial | 86.5% | 47.7% | 12.4% |
| $\ell_2$ adversarial | 86.3% | **48.4%** | **9.4%** |

and as the *fast gradient sign method* (FGSM) when the $\ell_\infty$ sign update is used [Goodfellow et al., 2015]. Note that in both cases the step size is $\alpha = \varepsilon$ which leads to perturbations located on the boundary of the set $\Delta$. These methods are fast but sometimes prone to *catastrophic overfitting* when the model overfits to FGM/FGSM but is not robust to iterative PGD attacks [Tramèr et al., 2018, Wong et al., 2020]. This problem can be alleviated by specific regularization methods like CURE [Moosavi-Dezfooli et al., 2019, Huang et al., 2020] or GradAlign [Andriushchenko and Flammarion, 2020]. However, for small enough $\varepsilon$, adversarial training with FGM/FGSM works as well as multi-step PGD [Andriushchenko and Flammarion, 2020].

**Experimental details.** We do experiments on two common image classification datasets: CIFAR-10 [Krizhevsky and Hinton, 2009] which has $32 \times 32$ images, and ImageNet-100 [Russakovsky et al., 2015] with $224 \times 224$ images where we take each tenth class following Laidlaw et al. [2021]. We choose ImageNet-100 since we always perform a grid search over the main hyperparameters such as the perturbation radius for adversarial training which would be too expensive to do on the full ImageNet. Unless mentioned otherwise, we use PreAct ResNet-18 architecture [He et al., 2016]. We specify the exact hyperparameters in App. A. We evaluate the accuracy on common corruptions using CIFAR-10-C and ImageNet-C datasets from [Hendrycks and Dietterich, 2019] which contain 15 different synthetic corruptions in 4 categories: blur, noise, digital, weather corruptions. We report the accuracy by averaging over all 5 severity levels.

**Adversarial training improves accuracy and calibration.** We start by showing in Fig. 1 the common corruption accuracy of $\ell_\infty$ adversarially trained models as it is the most widely studied setting [Madry et al., 2018] and has been reported multiple times in common corruption literature [Hendrycks et al., 2019b, Ford et al., 2019, Rusak et al., 2020]. Since we are interested primarily in small-$\varepsilon$ adversarial training, we rely throughout the paper on FGM/FGSM

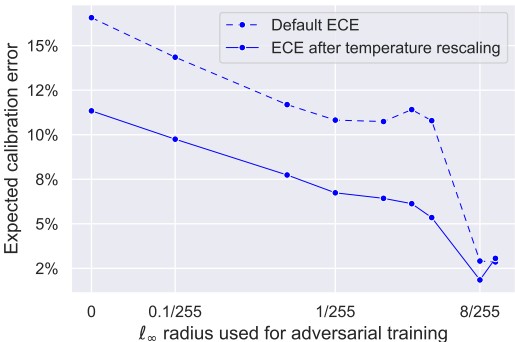

Figure 2: Expected calibration error on CIFAR-10-C for $\ell_\infty$ adversarially trained models.

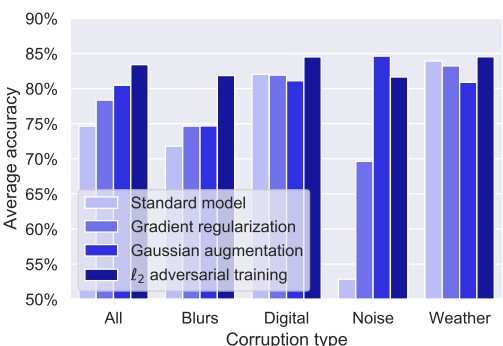

Figure 3: Accuracy for different corruption types on CIFAR-10-C. Unlike other methods, adversarial training improves the performance on each corruption.

for $\ell_2/\ell_\infty$ norms respectively to solve the inner maximization problem (1) which only leads to a $2\times$ computational overhead. Note however that we exceptionally use PGD with 10 steps for $\varepsilon \in \{8/255, 10/255\}$ to prevent catastrophic overfitting and allow a direct comparison with previous works. We observe that *for the small-$\varepsilon$ regime* around $\varepsilon = 1/255$, we get a significant improvement in corruption accuracy: 74.5% accuracy is achieved with standard training, 82.7% with adversarial training using $\varepsilon = 1/255$, and 73.8% using the standardly reported threshold $\varepsilon_\infty = 8/255$.[1] The reason is that the tradeoff between robustness and accuracy [Tsipras et al., 2019] has to be carefully balanced—if the standard accuracy drops for higher $\varepsilon$, the corruption accuracy also deteriorates. Thus, selecting the most robust $\ell_p$-model does not lead to the optimal performance on common corruptions. Alternatively, one can also balance this tradeoff by mixing clean and adversarial samples, but it overall leads to similar results (see App. C for details), so we focus on adversarial training with 100% adversarial samples for the rest of the paper.

Additionally, we show that predicted probabilities of adversarially trained models are significantly better *calibrated on common corruptions*. We believe that calibration is another important aspect of the model's trustworthiness, which is particularly important in the presence of out-of-distribution data such as corrupted images. In Fig. 2, we plot the expected calibration error (ECE) [Guo et al., 2017] on CIFAR-10-C for models trained with different $\ell_\infty$-radii. We observe that the ECE—both with and without temperature rescaling (see App. B for details)—follows a decreasing trend over $\ell_\infty$-radii which is expected since a classifier that predicts uniform probabilities over classes is perfectly calibrated. In particular, the most accurate model trained with $\varepsilon_\infty = 1/255$ has a much lower ECE than the standard model: 10.8% instead of 16.6%, and with temperature rescaling 6.7% instead of 11.3%.

We further compare the performance in the $\ell_2$ perturbation model. In Table 1, we report results of standard, $\ell_\infty$, and $\ell_2$ adversarial training on CIFAR-10 and ImageNet-100 where we perform a detailed grid search for each model over the perturbation radius $\varepsilon$. To the best of our knowledge, we show for the first time that adversarial training improves calibration (see also App. B) while increasing the accuracy and that it helps on ImageNet-C, and not only on CIFAR-10-C. We generally observe that $\ell_2$ adversarial training performs better than $\ell_\infty$, thus we focus on it in the next section.

# 4 UNDERSTANDING THE EFFECT OF ADVERSARIAL TRAINING ON IMAGE CORRUPTIONS

Here we compare $\ell_2$ adversarial training to other natural baselines and discuss the main conceptual differences.

**Comparing natural baselines across corruption types.** We compare $\ell_2$ adversarial training with a few simple baselines: standard training, gradient regularization [Drucker and LeCun, 1992], and standard Gaussian data augmentation. To ensure a fair comparison, we perform a grid search for each method over the perturbation radius $\varepsilon$, regularization parameter $\lambda$, and noise standard deviation $\sigma$ respectively. We choose to compare to gradient regularization since it is an established regularization method that may have a similar effect to adversarial training with small perturbations [Simon-Gabriel et al., 2019]. We aggregate the corruptions over each type (blurs, digital, noise, weather) and plot the results in Fig. 3 and report results over each corruption in Fig. 12 in the Appendix.

First, we observe that adversarial training is the best performing method and that unlike other methods, $\ell_2$ adversarial training helps for *each* corruption type. At the same time, Gaussian augmentation *degrades* the performance on digital

---

[1]The exact numbers differ from [Ford et al., 2019] since we use ResNet-18 instead of WRN-28-10 and different hyperparameters.

and weather corruptions while very significantly improving the performance for noise corruptions which is expected as the Gaussian noise used for training is also contained in the noise corruptions. Interestingly, for the fog and contrast corruptions, the performance degrades for *all* methods (see Table 10 in App. H), consistently with the observation made in Ford et al. [2019]. Our results also suggest that the impact of gradient regularization is limited and it cannot explain the accuracy gains of both adversarial training and Gaussian augmentation as one could expect from the fact that these methods are equivalent to gradient regularization when used with *sufficiently* small parameters $\sigma$ and $\varepsilon$ [Bishop, 1995].

**Worst-case vs average-case behavior.** Ford et al. [2019] show that the robustness to Gaussian noise and adversarial perturbations are closely related. More precisely, they show using concentration of measure arguments that a non-zero error rate under Gaussian perturbation implies the existence of small adversarial perturbations and consequently that improving adversarial robustness leads to an improvement in robustness against Gaussian perturbations. This finding is consistent with what we observe here. What remains to be understood is why adversarial training performs *better* than Gaussian augmentation on common corruptions. The main difference between both methods appears when analyzing the objectives that both methods minimize. For a single sample $x$, the loss function considered in Gaussian augmentation is:

$$\mathbb{E}_{d \sim N(0, I\sigma^2)} \left[ \ell(\theta, x + d) \right] \ \sim \ \mathbb{E}_{\rho : ||\rho||_2 = \sigma\sqrt{d}} \left[ \ell(\theta, x + \rho) \right],$$

since Gaussian vectors with variance $\sigma^2 I$ are highly concentrated on the sphere of radius $\sigma\sqrt{d}$ in high dimensions. Therefore Gaussian augmentation amounts to minimize an *averaged* objective where perturbations are averaged over the *sphere*. However, the objective behind adversarial training defined in Eq. (1) amounts to minimize a *worst-case* loss based on the worst-case perturbation in the *ball*. The key difference is that minimization of the expected value of the loss *does not guarantee* any behavior inside the sphere.

To investigate this behavior, we perform the following experiment in Fig. 4. For random 1000 test set images from CIFAR-10, we evaluate the loss with additive Gaussian noise of $\sigma \in [0, 0.1]$ and average the loss function over both images and perturbations for (1) a standard model, (2) a model trained with Gaussian augmentation with $\sigma = 0.05$ where all 100% training samples are augmented, (3) a model trained with Gaussian augmentation for $\sigma = 0.1$ where only 50% training samples are augmented, and (4) $\ell_2$ adversarially trained model with $\varepsilon = 0.1$. We notice that the loss function for 100% Gaussian augmentation is minimal at $\sigma$ which is only slightly less than $\sigma = 0.05$ used for its training. *Hence, the model has overfitted not only to the type of noise but also to its magnitude.* The loss function outside *and inside* of the sphere is bigger than on its surface. However, there is a simple fix if we train with 50%

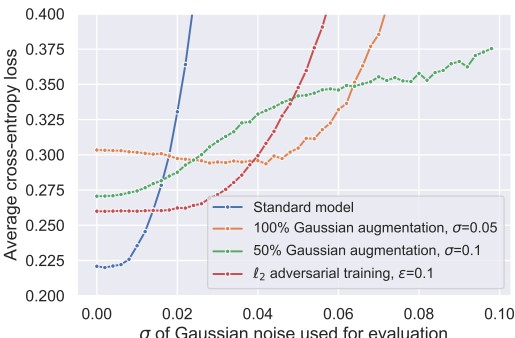

Figure 4: Average cross-entropy loss under Gaussian noise for different training methods.

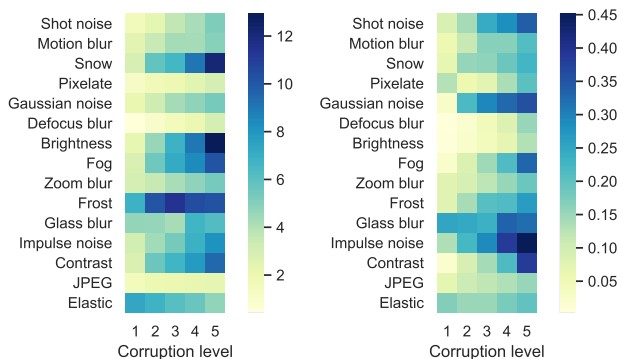

Figure 5: Average $\ell_2$ and LPIPS distance for different common corruptions from CIFAR-10-C.

Gaussian noise in each batch, as suggested, e.g., in Rusak et al. [2020] in contrast to Ford et al. [2019]. This scheme allows to alleviate the $\sigma$-overfitting behavior and also achieve better accuracy on clean samples (93.2% instead of 92.5%) and, most importantly, *significantly* improve on common corruptions (85.0% instead of 80.5%). At the same time, $\ell_2$ adversarial training does not suffer from this problem and both 100% and 50% schemes work nearly equally well (details can be found in App. C). We provide a further discussion on $\sigma$-overfitting in App. D together with additional experiments on ImageNet-100 where $\sigma$-overfitting has even more noticeable behavior.

**Local vs global $\ell_p$ behavior.** Interestingly, adversarial training with worst-case perturbations bounded within a *tiny $\ell_2$* ball leads to robustness significantly beyond this radius. Fig. 5 illustrates that common corruptions have an $\ell_2$ norm an *order of magnitude larger* than $\varepsilon = 0.1$ used for $\ell_2$ adversarial training. This is in contrast with adversarial robustness that does not significantly extend beyond the radius used for training [Madry et al., 2018]. Related to this, Ford et al. [2019] argue that *for Gaussian noise* improving the minimum distance to the decision boundary (e.g. via adversarial training) also leads to an improvement of the average distance. We have a similar mechanism at play for adversarial $\ell_2$ perturbations and common corruptions which may

explain the generalization of adversarial training to large average-case perturbations. However, our setting is more complex compared to Ford et al. [2019] since at the training and test time we deal with *different* and *diverse* types of noise.

# 5 IMPROVING ADVERSARIAL TRAINING BY RELAXING A PERCEPTUAL DISTANCE

As shown above, $\ell_p$ adversarial training already leads to encouraging results on common corruptions. Moreover, the $\ell_2$ distance appears to be more suitable for adversarial training than $\ell_\infty$ on both datasets as implied by Table 1. This observation suggests that using more advanced distances such as perceptual ones can further improve corruption robustness.

**From $\ell_p$ distances to LPIPS.** One of the main disadvantages of $\ell_p$-norms is that they are very sensitive under simple transformations such as rotations or translations [Sharif et al., 2018]. One possible solution is to consider *perceptual distances*[2] which capture these invariances better such as the *learned perceptual image patch similarity* (LPIPS) distance introduced in Zhang et al. [2018b] and which is based on the activations of a convolutional network. The LPIPS distance is formally defined as

$$\mathrm{d_{LPIPS}}(x, x')^2 = \sum_{l=1}^{L} \alpha_l \|\phi_l(x) - \phi_l(x')\|_2^2, \qquad (3)$$

where $L$ is the depth of the network, $\phi_l$ is its feature map up to the $l$-th layer, and $\{\alpha_l\}_{l=1}^{L}$ are some constants that weigh the contributions of the $\ell_2$ distances between activations. There are two crucial elements in LPIPS: the learned network and learned coefficients $\{\alpha_l\}_{l=1}^{L}$. Zhang et al. [2018b] propose to take a network pre-trained on ImageNet and learn coefficients on their collected dataset of human judgemenets about which images are closer to each other. Both Zhang et al. [2018b] and Laidlaw et al. [2021] argue about better suitability of LPIPS to measure image similarity. In App. E we analyse the suitability of LPIPS over $\ell_2$ specifically on the images from CIFAR-10-C with a detailed breakdown over corruption types. In particular, we show that the LPIPS distance is better correlated with the error rate of the network, and the increase over severity levels is more monotonic compared to $\ell_2$ as can be also seen in Fig. 5.

**LPIPS adversarial training.** In view of the positive features of LPIPS, adversarial training using LPIPS appears to be a promising approach to improve the performance on common corruptions. The worst-case loss problem considered in (1) using the LPIPS distance can be formulated

---

[2]Not necessarily distances in a strict mathematical sense that assumes a certain set of axioms to hold.

as:

$$\max_{\delta} \ell(x + \delta, y; \theta) \quad \text{s.t.} \quad \mathrm{d_{LPIPS}}(x, x + \delta) \le \varepsilon. \qquad (4)$$

However, this optimization problem is challenging since $\mathrm{d_{LPIPS}}$ is itself defined by a neural network, and the projection onto the LPIPS-ball—as required when using PGD to solve (4)—does not admit a closed-form expression. This problem was considered in Laidlaw et al. [2021] who propose two approximate attacks: the Perceptual Projected Gradient Descent (PPGD) and the Lagrangian Perceptual Attack (LPA). We discuss their approach in more detail in App. F but emphasize that they either need to perform an approximate projection which is computationally expensive or come up with some scheme for tuning the Lagrange multiplier $\lambda$ in the Lagrangian formulation. Furthermore, they suggest in both cases to use 10-step iterative attacks for approximate LPIPS adversarial training which limits the scalability of the method to large datasets such as ImageNet.

**Relaxed LPIPS adversarial training.** We propose here a relaxation of the LPIPS adversarial objective (4). For the simplicity of presentation, let us start by assuming that the LPIPS distance is defined using a *single* intermediate layer of the network, i.e. $\mathrm{d_{LPIPS}}(x, x') = \|\phi(x) - \phi(x')\|_2$. Then we can write a neural network $f$ as the composition of the feature map $\phi$ and the remaining part of the network $f(x) = h(\phi(x))$. The LPIPS adversarial objective (4) in this notation becomes

$$\max_{\delta} \ell(h(\phi(x + \delta))) \quad \text{s.t.} \quad \|\phi(x + \delta) - \phi(x)\|_2 \le \varepsilon.$$

We first introduce the slack variable $\tilde{\delta} = \phi(x + \delta) - \phi(x)$ which allows us to rewrite the objective as

$$\max_{\delta, \tilde{\delta}} \ell(h(\phi(x) + \tilde{\delta})) \ \text{s.t.} \ \|\tilde{\delta}\|_2 \le \varepsilon, \ \tilde{\delta} = \phi(x + \delta) - \phi(x).$$

Then we perform the key step: we omit the constraint on the slack variable and obtain the following relaxation

$$\max_{\tilde{\delta}} \ell(h(\phi(x) + \tilde{\delta})) \quad \text{s.t.} \quad \|\tilde{\delta}\|_2 \le \varepsilon, \qquad (5)$$

i.e. we lift the requirement that there should exist a $\delta$ in the *input* space that corresponds to the layerwise perturbation $\tilde{\delta}$.

A similar relaxation can be derived when the LPIPS distance is defined using multiple layers (see App. F):

$$\max_{\tilde{\delta}^{(1)}, \dots, \tilde{\delta}^{(L)}} \ell(g_L(\dots g_1(x + \tilde{\delta}^{(1)}) \dots + \tilde{\delta}^{(L)})) \qquad (6)$$

$$\text{s.t.} \quad \|\tilde{\delta}^{(l)}\|_2 \le \varepsilon_l \ \forall l \in \mathcal{L}_{LPIPS}, \quad \tilde{\delta}^{(l)} = 0 \ \forall l \notin \mathcal{L}_{LPIPS},$$

where the network is written under its compositional form $f = g_L \circ \dots \circ g_1$, $\mathcal{L}_{LPIPS}$ is the set of layer indices used in LPIPS and $\varepsilon_l$ denotes the $\ell_2$ bound imposed at the $l$-th layer. We denote this relaxation as *relaxed LPIPS adversarial training* (RLAT) and solve it efficiently using a single-iteration adversarial attack similar to FGM. We emphasize

that the projection of each $\tilde{\delta}^{(l)}$ onto the corresponding $\ell_2$ balls is computationally cheap to perform, unlike the LPIPS projection.

Since we perform relaxation and *train* the network which is also used to compute LPIPS, the exact layerwise coefficients $\alpha_l$ from the original LPIPS Zhang et al. [2018b] are no longer applicable and cannot be used to set the layerwise bounds $\varepsilon_l$. Therefore, we set our own values of $\varepsilon_l$ which we specify in App. F together with detailed derivations of RLAT, its precise algorithm and other implementation details. Finally, we remark that related layerwise adversarial training methods have been proposed before [Stutz et al., 2019, Volpi et al., 2018, Wei and Ma, 2020]. However, viewing layerwise adversarial training as an efficient relaxation of LPIPS adversarial training is novel, as well as applying these methods for general robustness such as common corruptions.

# 6 EMPIRICAL EVALUATION OF RLAT

Here we first show that RLAT indeed substantially improves the LPIPS robustness. Second, we compare RLAT to other established methods and show that it consistently leads to improved accuracy and calibration on common corruptions.

**LPIPS robustness of RLAT.** We use the Lagrangian Perceptual Attack attack developed in Laidlaw et al. [2021] to estimate the LPIPS adversarial accuracy under different LPIPS radii and plot results in Fig. 6 on CIFAR-10. We use standard, $\ell_2$ adversarial training (AT), Fast PAT, and RLAT models with their main hyperparameters selected to perform best on common corruptions.[3] We observe that RLAT indeed substantially improves LPIPS robustness, even more than other approaches such as $\ell_2$ AT and Fast PAT. This gives further evidence that both $\ell_2$ and RLAT training *do not suffer from catastrophic overfitting*, even though trained with one-step perturbations similar to FGSM. We provide a similar evaluation for $\ell_2$ robustness in App. F (Fig. 10).

**Main experimental setup.** We compare the results for RLAT with additional baselines: $\ell_2$ and $\ell_\infty$ adversarial training (with 100% adversarial samples per batch), Gaussian augmentation (with both 50% and 100% augmentations per batch), AdvProp [Xie et al., 2020], Fast PAT [Laidlaw et al., 2021], and also four data augmentation approaches: DeepAugment [Hendrycks et al., 2021], AugMix [Hendrycks et al., 2019b], adversarial noise training (ANT) [Rusak et al., 2020], and Stylized ImageNet (SIN) [Geirhos et al., 2019]. We use AugMix method additionally with the Jensen-

---

[3]We note that Laidlaw et al. [2021] focus on robustness to unseen adversarial examples that involve a *worst-case* optimization process, while we focus on unseen *average-case* common corruptions. This is the reason why the optimal perturbation radii that we consider are noticeably smaller than in their paper.

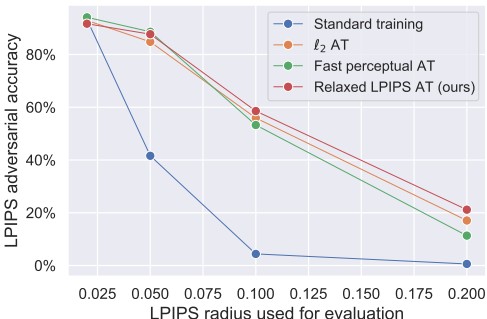

Figure 6: LPIPS adversarial robustness of different training schemes on CIFAR-10.

Shannon regularization term as proposed in Hendrycks et al. [2019b]. We train all methods from random initialization except ANT where we follow the scheme of Rusak et al. [2020]. All comparisons between methods are performed with a grid search over their main hyperparameters (reported in App. A) such as $\sigma$ in Gaussian augmentation or $\epsilon$ in adversarial training which we perform on the main 15 corruptions from CIFAR-10-C / ImageNet-C. In App. H we further verify that selecting the main hyperparameters on validation corruptions leads to the same results. For Fast PAT on CIFAR-10, we do a grid search over their parameter $\varepsilon$, but on ImageNet-100 we report the results based on the models provided by the authors due to limited computational resources. To assess calibration, we report the expected calibration error (ECE) (see App. H for ECE with temperature rescaling Guo et al. [2017]). More details can be found in our repository `https://github.com/tml-epfl/adv-training-corruptions`.

Since the main goal of the common corruption benchmark [Hendrycks and Dietterich, 2019] is to show the model's behavior on *unseen* corruptions, we do not use overlapping augmentations in training (see App. A). The only exception is Gaussian augmentation which we mark in gray in Table 2 following [Rusak et al., 2020] since it belongs to common corruptions. We note that removing only Gaussian noise from evaluation is not sufficient, because other noises can be affected as well by training with Gaussian augmentation. Thus, the results of 100% and 50% Gaussian augmentation are shown only for illustrative purposes suggesting that adversarial training with no prior knowledge about the corruptions can obtain almost the same results as direct augmentation.

**Main experimental results.** We show the main experimental results on CIFAR-10-C and ImageNet-100-C in Table 2. First of all, we observe that $\ell_p$ adversarial training is a strong baseline on common corruptions on both datasets with a larger gain on CIFAR-10-C. Using our proposed relaxed LPIPS adversarial training further improves the corruption accuracy on both datasets: from 74.6% to 84.1% on CIFAR-

Table 2: Accuracy and calibration of ResNet-18 models trained on CIFAR-10 and ImageNet-100. Gray-colored numbers correspond to methods partially trained with the corruptions from CIFAR-10-C and ImageNet-100-C.

| Training | Standard accuracy | Corruption accuracy | Corruption calibr. error |
|---|---|---|---|
| **CIFAR-10** | | | |
| Standard | 95.1% | 74.6% | 16.6% |
| 100% Gaussian | 92.5% | 80.5% | 13.2% |
| 50% Gaussian | 93.2% | 85.0% | 9.1% |
| Fast PAT | 93.4% | 80.6% | 12.0% |
| AdvProp | 94.7% | 82.9% | 10.1% |
| $\ell_\infty$ adversarial | 93.3% | 82.7% | 10.8% |
| $\ell_2$ adversarial | 93.6% | 83.4% | 10.5% |
| RLAT | 93.1% | **84.1%** | **9.9%** |
| DeepAugment | 94.1% | 85.3% | 8.7% |
| DeepAugment + RLAT | 93.6% | **87.8%** | **6.1%** |
| AugMix | 95.0% | 86.6% | 6.9% |
| AugMix + RLAT | 94.8% | **88.5%** | **4.5%** |
| AugMix + JSD | 95.0% | 88.6% | 6.5% |
| AugMix + JSD + RLAT | 94.8% | **89.6%** | **5.4%** |
| **ImageNet-100** | | | |
| Standard | 86.6% | 47.5% | 10.0% |
| 100% Gaussian | 86.4% | 46.7% | 11.7% |
| 50% Gaussian | 83.8% | 55.2% | 6.1% |
| Fast PAT | 71.5% | 45.2% | 8.0% |
| $\ell_\infty$ adversarial | 86.5% | 47.7% | 12.4% |
| $\ell_2$ adversarial | 86.3% | 48.4% | 9.4% |
| RLAT | 86.5% | **48.8%** | **9.1%** |
| AugMix | 86.7% | 52.3% | 7.5% |
| AugMix + RLAT | 86.8% | **54.8%** | **4.7%** |
| AugMix + JSD | 88.4% | 59.3% | 1.9% |
| AugMix + JSD + RLAT | 87.1% | **61.1%** | **1.8%** |
| SIN | 86.6% | 53.7% | 6.7% |
| SIN + RLAT | 86.5% | **54.3%** | **6.0%** |
| ANT$^{3x3}$ | 85.9% | 57.7% | 5.1% |
| ANT$^{3x3}$ + RLAT | 85.3% | **58.3%** | **4.4%** |

10-C and from 47.5% to 48.8% compared to standard models. Moreover, RLAT also improves calibration compared to the standard model: from 16.6% to 9.9% ECE on CIFAR-10-C and from 10.0% to 9.1% ECE on ImageNet-100-C. We also observe that 100% Gaussian augmentation even deteriorates the performance on ImageNet-100-C while 50% Gaussian augmentation significantly improves the average accuracy which is consistent with Rusak et al. [2020].

We observe that RLAT can be successfully combined with existing data augmentations, leading to better accuracy and calibration. E.g., adding RLAT on top of DeepAugment helps to improve the CIFAR-10-C accuracy from 85.3%

Table 3: Wall-clock time in hours for ResNet-18 trained with different methods on CIFAR-10 and ImageNet-100 using one Nvidia V100 GPU. [*] denotes the time reported by Laidlaw et al. [2021] for a larger model (ResNet-50) using different hardware (4 Nvidia RTX 2080 Ti GPUs).

| Training | Dataset | |
|---|---|---|
| | CIFAR-10 | ImageNet-100 |
| Standard | 0.8h | 3.9h |
| $\ell_2/\ell_\infty$ adversarial | 1.3h | 5.8h |
| RLAT | 1.8h | 6.2h |
| Fast PAT | 9.4h | [*]120h |

to 87.8%. Combining RLAT with the AugMix augmentation improves the corruption accuracy from 86.6% to 88.5% on CIFAR-10-C and on ImageNet-100-C from 52.3% to 54.8%. Combining SIN and ANT$^{3x3}$ improves the accuracy on ImageNet-100-C from 53.7% to 54.3% and from 57.7% to 58.3%, respectively. Moreover, we see that RLAT consistently improves ECE in all settings, and we refer to App. H for ECE with temperature rescaling which qualitatively shows the same behavior.

Additionally, we added our models to the RobustBench leaderboard[4] where our method has the best performance among the architectures of comparable sizes (i.e., ResNet-18). The models which perform better have larger architectures and some of them additionally rely on ensembles.

**Runtime of RLAT.** We report a full runtime comparison between standard training, $\ell_2$ / $\ell_\infty$ adversarial training, RLAT, and Fast PAT in Table 3. The main observation is that RLAT is significantly faster than Fast PAT (e.g., 1.8 hours vs. 9.4 hours on CIFAR-10) and leads only to a slight overhead compared to $\ell_2$ / $\ell_\infty$ adversarial training (1.8 hours vs 1.3 hours on CIFAR-10). These runtimes show further the advantage of the single-step adversarial training procedure of RLAT compared to the multi-step approach of Fast PAT. It would be interesting in future work to develop a single-step version of Fast-LPA which is, however, not straightforward because of their Lagrangian formulation and the need to tune the parameter $\lambda$ over the iterations of Fast-LPA.

**Additional experiments.** We refer to the Appendix for further experimental results. In App. G, we evaluate the performance of the models from Table 2 on ImageNet-A, ImageNet-R, and Stylized ImageNet to better understand how well the improvements on common corruptions transfer to other distribution shifts. In App. H, we provide more detailed results such as those presented in Table 2 but with breakdowns over different corruptions and severities. We also present results for larger network architectures and for AugMix combined with $\ell_p$ adversarial training in App. H, as well as results of RLAT over multiple random seeds.

---

[4]https://robustbench.github.io/

# 7 CONCLUSIONS AND FUTURE WORK

Our findings suggest that adversarial training can be successfully used to improve accuracy and calibration on common image corruptions. Even simple $\ell_p$ adversarial training can serve as a strong baseline if the optimal perturbation radius is chosen for the given problem. More advanced adversarial training schemes involve perceptual distances, such as LPIPS, and we provide a relaxation of LPIPS adversarial training with an efficient single-step procedure. We observe that the developed relaxation (RLAT) substantially improves the LPIPS robustness and can be successfully combined with existing data augmentations. We hope that RLAT would be of interest also for other domains such as natural language processing where robustness to commonly occurring corruptions (e.g., typos) is an important task.

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
