# OpenReview forum: "On the Effectiveness of Adversarial Training Against Common Corruptions"
_auai.org/UAI/2022/Conference — UAI 2022 Poster_

### Official Review · Reviewer_wAsH · 2022-04-10

**Q2(1) Originality/Novelty:** 3
**Q2(2) Significance/Impact:** 2
**Q2(3) Correctness/Technical Quality:** 2
**Q2(6) Clarity Of Writing:** 3
**Q6 Overall Score:** 5
**Q8 Confidence In Your Score:** 3

**Q1 Summary And Contributions:**

In this paper, the effect of adversarial training on common corruptions is discussed from theoretical and  experimental aspects. Then a new adversarial training is proposed to obtain very good performance on common corrputions.

**Q2 Assessment Of The Paper:**

More detailed information regarding each of these aspects is given below:

**Q2(4) Quality Of Experiments (Optional):**

2: Fair: The experimental evaluation is weak: important baselines are missing, or the results do not adequately support the main claims.

**Q2(5) Reproducibility:**

2: Fair: Key resources (e.g., proofs, code, data) are unavailable but key details (e.g., proof sketches, experimental setup) are sufficiently well-described for an expert to confidently reproduce the main results.

**Q3 Main Strengths:**

+ the link between adversarial examples and normal corruptions is interesting
+ if can be proved that adversarial training is helpful to normal corruptions, then it can make AT more useful.

**Q4 Main Weakness:**

- the claim is not supported by theoretical analysis but is based on numerical experiments.
- there are many unclear parts for adversarial training, which can be found in Q5.
- the newly proposed LPIPS is not a promising new AT method, and to some degree, is not necessary for the main claim, i.e., AT can help normal corruptions, since this main claim is for all ATs. It is better to verify standard and commonly used AT, not a new one, to verify the claim.

**Q5 Detailed Comments To The Authors:**

- in Table 2, the natural accuracy of adversarial training reported in this paper is unexpectedly high. Is that because ATs in this paper stop in a very early stage? In that case, the link of AT and normal corruption will be weaken.
- to ensure the model is sufficient trained by adversarial examples, it is better to report robust accuracy, i.e., accuracy under attack. It is not necessary to conduct SOTA attacks, but it is suggested to tell us the accuracy under AutoAttack, which is also easy for the authors to implement.
- when a model is sufficient trained by adversarial example sufficiently, its ECE is usually increased largely. But it is not the case in Table 2, where the ECE is even lower than standard training. Maybe I have some misunderstood?
- how about the experiment detail for Fig.1 ?

**Q7 Justification For Your Score:**

The topic is interesting. If the link between AT and noise corruption is verified, it can help the understand of adversarial examples and normal corruptions, and also make AT more useful. However, there are some unclear parts in experiments.

**Q9 Complying With Reviewing Instructions:**

1: Yes.

---

### Official Review · Reviewer_ym3L · 2022-04-11

**Q2(1) Originality/Novelty:** 2
**Q2(2) Significance/Impact:** 2
**Q2(3) Correctness/Technical Quality:** 3
**Q2(6) Clarity Of Writing:** 2
**Q6 Overall Score:** 5
**Q8 Confidence In Your Score:** 4

**Q1 Summary And Contributions:**

This paper focuses on how to improve adversarial training against common corruption. The authors find l_p adversarial training with an appropriately selected perturbation radius can serve a strong baseline for common corruption. However, it may overfit the perturbation size. Thus, this paper proposes relaxed LPIPS adversarial training, which shows promising results on improving the model performance.

**Q2 Assessment Of The Paper:**

More detailed information regarding each of these aspects is given below:

**Q2(4) Quality Of Experiments (Optional):**

3: Good: The experimental evaluation is adequate, and the results convincingly support the main claims.

**Q2(5) Reproducibility:**

3: Good: Key resources (e.g., proofs, code, data) are available and key details (e.g., proofs, experimental setup) are sufficiently well-described for competent researchers to confidently reproduce the main results.

**Q3 Main Strengths:**

1.	This paper is clearly written. The problem motivation and experiments are clearly shown.
2.	The current proposed method is technically sound and the corresponding properties is comprehensively understood by the various experiments.


**Q4 Main Weakness:**

1.	The current motivation of adopting LPIPS is unclear and not strong for this problem.
2.	The relationship of current method with the common corruption is unclear.
3.	The optimization problem of adversarial training has not been discussed in this draft. The detailed reason for this part has listed in Q5.
4.	The claim “leading to the state-of-the-art performance on common corruptions” is too strong and not well justified by current experiments.


**Q5 Detailed Comments To The Authors:**

1.	For the claim “leading to the state-of-the-art performance on common corruptions”, could the author provide the direct evidence through the evaluation from “robustbench” (which is a well-known benchmark) to strongly support this claim? To well justify this point, you are strongly encouraged to also provide the results in CIFAR-100-C, or provide some discussion about the relationship between the performance gain with the class numbers.
2.	It is already known that adversarial training with large epsilon ball will also cause the training instability, the corresponding loss landscape is hard to optimize [1], and has detrimental effects on the model accuracy and robustness. So, whether this is also one of the main causes for the observation of this paper? The high-level idea of the proposed relaxed LPIPS adversarial training seems to be consistent with previous maximization relaxation or curriculum strategies for improving adversarial training. What is the unique point of the proposed RLAT (especially for those work targeted in curriculum adversarial training e.g., [2])? Why do we need to choose LPIPS distance as the measurement? Is there any close relationship between LPIPS with the common corruption? The current presentation can hardly find a reasonable motivation for this part. If the uniqueness of the proposed RLAT and the clear motivation can be stated, the presentation quality can be further improved.
[1]. Liu, Chen, et al. "On the loss landscape of adversarial training: Identifying challenges and how to overcome them." Advances in Neural Information Processing Systems 33 (2020): 21476-21487.
[2]. Cai, Qi-Zhi, Chang Liu, and Dawn Song. "Curriculum Adversarial Training." IJCAI. 2018.


**Q7 Justification For Your Score:**

To sum up, the current version of this work lacks in-depth insights into the necessities of the proposed method and the discussion of the well-known properties (hard to optimize) of adversarial training, as well as stronger support for the “strong” claim (leading to the new state of the arts). Hence, it leaning my decision towards weak rejection.


After reading the rebuttal, I feel satisfied and increase my scores.

**Q9 Complying With Reviewing Instructions:**

1: Yes.

---

### Official Review · Reviewer_4ztG · 2022-04-15

**Q2(1) Originality/Novelty:** 3
**Q2(2) Significance/Impact:** 3
**Q2(3) Correctness/Technical Quality:** 3
**Q2(6) Clarity Of Writing:** 4
**Q6 Overall Score:** 7
**Q8 Confidence In Your Score:** 4

**Q1 Summary And Contributions:**

This paper empirically observes that adversarial training (AT), which proper radius selection, $\ell_p$ norm selection, and one-step FGM approximation, can be effective against common corruptions, whereas Gaussian augmented training cannot due to the thin-shell concentration problem of Gaussian distribution. Therefore, the paper is motivated to improve raidus/threat model selection of AT to defend against common corruptions, which results in RLAT approach, achieving SOTA on CIFAR-10-/ImageNet-C.

**Q2 Assessment Of The Paper:**

More detailed information regarding each of these aspects is given below:

**Q2(4) Quality Of Experiments (Optional):**

4: Excellent: The experimental evaluation is comprehensive and the results are compelling.

**Q2(5) Reproducibility:**

2: Fair: Key resources (e.g., proofs, code, data) are unavailable but key details (e.g., proof sketches, experimental setup) are sufficiently well-described for an expert to confidently reproduce the main results.

**Q3 Main Strengths:**

- Novel empirical findings about adversarial training and Gaussian augmented training on their generalizability for defending against common corruptions.
- Novel RLAT training approach with clear motivation as a generalized form of adversarial training.
- Extensive experimental evaluation supporting the claims.
- Good presentation quality.

**Q4 Main Weakness:**

- Since adversarial training is a bit tricky in practice, i.e., the selection of update steps/update sizes/random initialization/etc drastically affects the resulting robustness, I would strongly encourage the authors to open-source the code or present the key hyperparameters with pseudocode in the paper. In the current form, the key hyperparameters are scattered across main text and appendices which makes it hard to fully reproduce the results.
- The performance of RLAT is not that significant compared with $\ell_p$ AT training. I understand that $\ell_p$ AT training is a strong baseline as the paper justified. But the improvement is still a bit marginal and cannot outperform AugMix if used alone.

**Q5 Detailed Comments To The Authors:**

See main weaknesses.

Minor:

1. Since Gaussian augmentation has the noise concentration problem, we can consider other noise distribution that overcomes the problem as an evaluation baseline such as the distribution mentioned in [a].

[a] Zhang, Dinghuai, et al. "Black-box certification with randomized smoothing: A functional optimization based framework." Advances in Neural Information Processing Systems 33 (2020): 2316-2326.

2. On page 6 second column: $\delta = \phi(x) − \phi(x + \delta)$ seems that the two terms should be swapped.

**Q7 Justification For Your Score:**

The paper studies an important and practical problem and provides novel insights. Based on the novel insights, they propose a novel approach that is shown effective for training robust models against common corruptions. Though there are some weaknesses, I think the paper is clearly above the acceptance threshold.

**Q9 Complying With Reviewing Instructions:**

1: Yes.

---

### Decision · Program_Chairs · 2022-05-15

**Decision:**

Accept (Poster)

**Comment:**

Meta Review: Thank you for your submission to UAI and for your detailed response to the reviewers' concerns.

Reviewers appreciated the importance of adversarial training for improving robustness against common corruptions.  Reviewers found the RLAT training approach novel and the empirical evaluation to be comprehensive and extensive.   We encourage the authors to include the additional details and revisions (links to source code, RobustBench evaluation, additional citations, etc.) mentioned in the discussion in updates.